# Passive surveillance of human African trypanosomiasis in the Democratic Republic of the Congo: clinical presentation and prospective evaluation of rapid diagnostic and reference laboratory test accuracy

Jacquies Makabuza[1], Ipos Ngay Lukusa[2], Crispin Lumbala[1,3], Erick Mwamba Miaka[1], Pathou Nganzobo[1], Alain Fukinsia[1], Jean Kwete[1], Nicolas Bebronne[4], Philippe Büscher[4], Dieudonné Mumba Ngoyi[2], Veerle Lejon [5]*

1 Programme National de Lutte contre la Trypanosomiase Humaine Africaine, Kinshasa, Congo, 2 Institut National de Recherche Biomédicale, Kinshasa, Congo, 3 Université Officielle de Mbujimayi, Kasai Oriental, Congo, 4 Institute of Tropical Medicine, Antwerpen, Belgium, 5 Institut de Recherche pour le Développement, CIRAD, University of Montpellier, Montpellier, France

* veerle.lejon@ird.fr

## Abstract

### Background

Passive screening of gambiense human African trypanosomiasis (HAT) is based on rapid diagnostic tests (RDT), but sensitivity of the currently commercialised RDTs has hardly been assessed prospectively. In view of the increasing importance of remote testing for HAT, the diagnostic performance of reference laboratory tests also needs further documentation.

### Methodology/Principal Findings

The study is registered in ClinicalTrials.Gov under identifier NCT03356665. Clinical suspects in 29 health facilities in DR Congo were screened consecutively between October 2017 and December 2020 with 3 HAT RDTs, including HAT Sero *K*-SeT, an RDT that is nowadays still commercialised. HAT RDT positives were examined parasitologically and their dried blood spots tested in trypanolysis, indirect ELISA/*T.b. gambiense*, LAMP *Trypanosoma brucei* Detection Kit and m18S and TgsGp qPCR. Association of clinical signs with HAT, and sensitivity, specificity, and predictive values of the screening and reference laboratory tests were estimated using parasitology as the gold standard. Trypanosomes were detected in 42/3113 study participants. Logistic regression revealed that sleep disruption, enlarged lymph nodes, psychiatric problems, recurrent fever not responding to anti-malarials and motor disorders were significantly associated with HAT (*p* < 0.05, odds 3.0-10.6). Together, the RDTs detected 253/3113 seropositives. Sensitivity and specificity of HAT Sero *K*-SeT were

**Data availability statement:** Metadata, the trial protocol and Standard Operating Procedures of all rapid diagnostic tests and reference laboratory tests are freely available under a CC-BY license via https://doi.org/10.23708/LTTOWL . However, some of the individual-level data files remain under restricted access to ensure compliance with European and French data protection laws (GDPR) and the ethical guidelines approved by the IRD institutional review board. These data include potentially sensitive information that could compromise participant confidentiality. The data sharing plan grants access to the restricted files, upon request via data@ird.fr, and, subject to the signing of a data sharing agreement. This process ensures both legal compliance and the protection of personal data.

**Funding:** This study was funded by the EDCTP2 programme supported by the European Union (grant number DRIA-2014-306-DiTECT-HAT, VL). The funders had no role in study design, data collection and analysis, decision to publish, or preparation of the manuscript.

**Competing interests:** The authors have declared that no competing interests exist.

respectively 100% (42/42; 95% CI 91.6-100%) and 93.9% (2882/3071; 95% CI 92.9-94.6%). Specificities of the reference laboratory tests were ≥ 91.6%, except for LAMP. Sensitivity of ELISA/*T.b. gambiense* and trypanolysis were 93.9% (31/33; 95% CI 80.4-98.9) and 84.9% (28/33; 95% CI 69.1-93.4), and were ≤ 63.6% for LAMP, m18S and TgsGp qPCR.

## Conclusions/Significance

Compared to the WHO's target product profiles for gambiense HAT RDTs, the HAT Sero *K*-SeT RDT had ideal sensitivity but its specificity was on the borderline of minimally acceptable. Sub-optimal sensitivities of trypanolysis and to a lesser extent, indirect ELISA/*T.b. gambiense* when applied on DBS, were confirmed. Molecular tests for remote testing need to be improved and evaluated further.

## Author summary

While gambiense human African trypanosomiasis (HAT) approaches elimination, its diagnosis increasingly relies on fixed health structures screening clinical suspects with rapid diagnostic tests (RDT). For confirmation of infection and for epidemiological purposes, remote testing of blood specimens from RDT seropositives also gains importance. The low HAT prevalence hampers assessment of the sensitivity of HAT RDTs and remote tests. All available results should therefore be published. We prospectively evaluated HAT RDTs and remote reference laboratory tests diagnostic performances in passive screening in 3 countries, and report the results of the Democratic Republic of the Congo here. The sensitivity of the HAT Sero *K*-SeT, one of the 2 HAT RDTs that are actually commercialised, complied with the ideal >99% sensitivity of the target product profile established by WHO, but its specificity was on the limit of minimally acceptable, set by WHO at >95%. Antibody detection with trypanolysis and/or indirect ELISA/*T.b. gambiense* is already routinely implemented, but the use of dried blood spots seems to reduce the tests' sensitivity. LAMP *Trypanosoma brucei* Detection Kit, m18S and TgsGp qPCR were not reliable. These results confirm other study data from Guinea and Côte d'Ivoire. The diagnostic performance of newly emerging molecular tests, should be further evaluated.

## Introduction

Screening for presence of specific antibodies is a crucial step in the diagnosis and control of gambiense human African trypanosomiasis (HAT), a tropical neglected disease due to *Trypanosoma brucei* (*T.b.*) *gambiense*. Infection with this parasite causes a fatal disease occurring in West and Central Africa, which, in its initial stage, is accompanied by non-specific clinical symptoms and signs, but elicits a strong

antibody response. Positivity in serological screening is a first step towards microscopic confirmation and live saving treatment in case trypanosomes are detected [1,2]. Moreover, with the introduction of new effective and safe oral drugs [3,4], widened treatment of seropositives, without mandatory parasitological confirmation, also called a "screen and treat" strategy, is increasingly considered [5,6].

The introduction of rapid diagnostic tests (RDT) for serological screening has been a major step forward towards elimination of gambiense HAT [7,8]. Such RDTs are easy to perform and fully adapted for use in primary health care settings that perform passive screening. The first-generation HAT RDTs, HAT Sero *K*-SeT and SD Bioline HAT were based on native *T.b. gambiense* variable surface glycoproteins (VSG) [7,8]. Second generation HAT RDTs consist of a combination of recombinant VSG with recombinant invariable surface glycoprotein 65 (rHAT Sero-Strip, Abbott Bioline HAT 2.0) [9,10]. At present, only HAT Sero *K*-SeT and Abbott Bioline HAT 2.0 are commercialised. Different prospective RDT performance evaluations have taken place, but due to the progress in gambiense HAT elimination [11], few HAT patients can be included and reported sensitivities are based on relatively small numbers [10,12–15]. Considering the limited data on HAT RDT sensitivity, all available evidence should be published. In addition, the diagnostic performance, and in particular sensitivity, of remote reference laboratory tests for HAT also needs to be documented in view of their growing importance for country verification of elimination and to follow up HAT epidemiology once "screen and treat" is introduced [6,16].

We here report on a prospective diagnostic performance evaluation of 3 HAT RDTs, HAT Sero *K*-SeT, SD Bioline HAT and rHAT Sero-Strip, during passive screening in the Democratic Republic of the Congo (DR Congo). This evaluation in DR Congo, the country reporting three quarters of all gambiense HAT cases, was part of the multi-country DiTECT-HAT-WP2 diagnostic study (NCT03356665) of which the results obtained in Côte d'Ivoire and Guinea have been published previously [12,13]. Taking into account that the SD Bioline HAT and rHAT Sero-Strip RDTs used in the study are not available anymore, we will focus on diagnostic performance of clinical symptoms and signs, of HAT Sero *K*-SeT, and of the remote reference laboratory tests performed on dried blood spots (DBS).

## Materials and methods

### Ethics considerations

The multi-country diagnostic trial "Diagnostic tools for human African trypanosomiasis elimination and clinical trials work package 2, passive case detection" (DiTECT-HAT-WP2) was registered on ClinicalTrials.Gov under identifier NCT03356665. Before its initiation, it received ethical clearance from the Advisory Committee on Deontology and Ethics of the French National Institute for Research on Sustainable Development (plenary meeting of 17–20 October 2016), the Institutional Review Board of the Institute of Tropical Medicine in Antwerp, Belgium (reference 1133/16), and the Ethics Committee of the University of Antwerp (Belgian registration number B300201730927). In DR Congo, DiTECT-HAT-WP2 was approved by the Comité d'Éthique de l'École de Santé Publique of Kinshasa University. Potential study participants were informed about the study, and gave written informed consent before inclusion. For minor participants, an assent was obtained and written informed consent was provided by the parents or legal guardians. The study was carried out according to the Declaration of Helsinki.

### Study setting

Study participants in DR Congo were recruited prospectively and consecutively between October 2017 and December 2020 among HAT clinical suspects presenting in 29 health facilities in the provinces of Kinshasa and Kwilu (Fig 1). In these 2 provinces, the HAT prevalence in 2017 was respectively 2.92, and 1.51 per 10,000 inhabitants, and it decreased to 0.97 and 0.99 per 10,000 inhabitants in 2019. Twenty-one serological screening sites (SSS) offered clinical and serological screening for HAT, and referred RDT positives to a centre for diagnosis and treatment (CDT). Eight CDTs did clinical and serological screening and performed parasitological confirmation. The SSS in Kinshasa province were the health

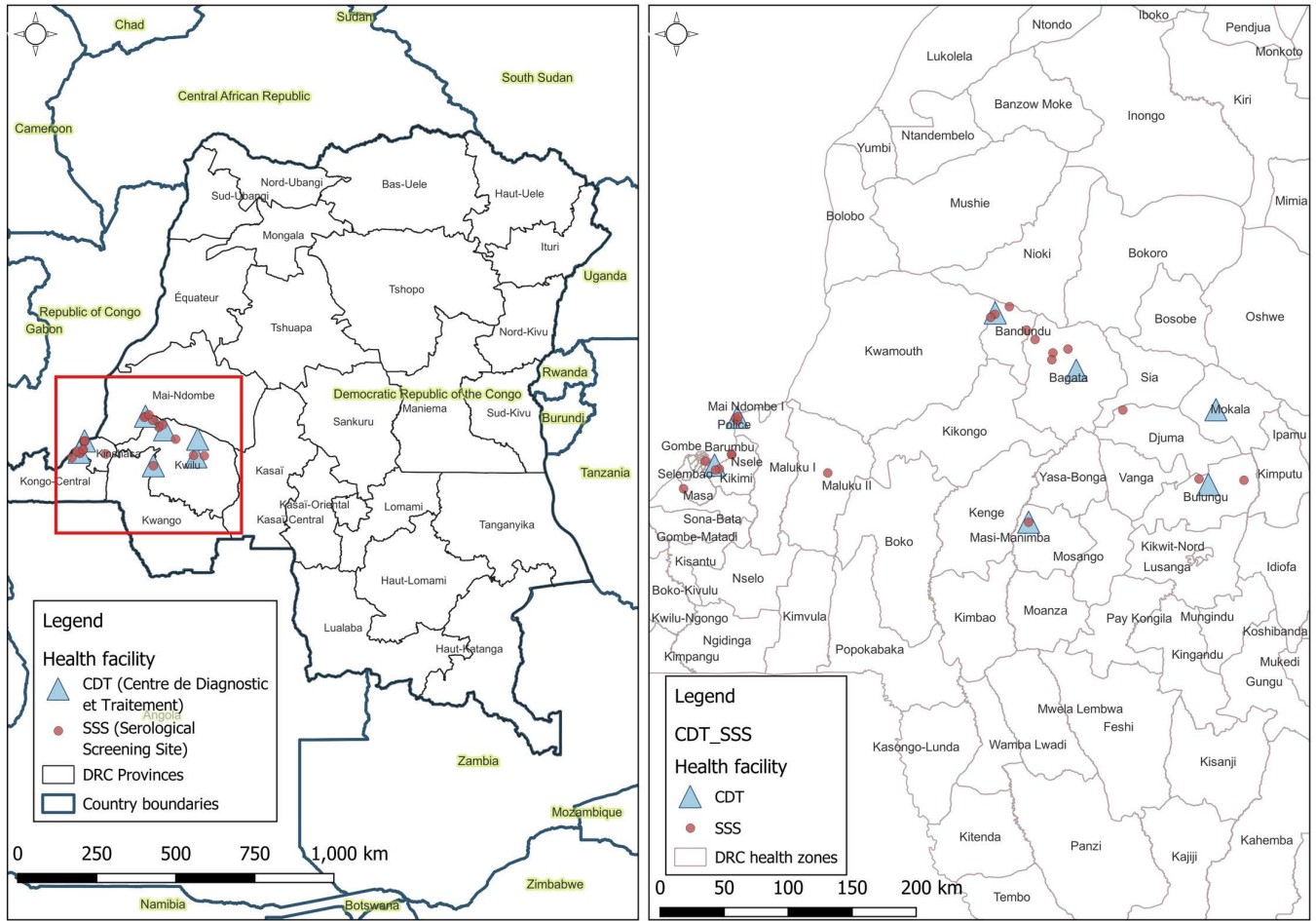

**Fig 1. Geographic localisation of serological screening sites and centres for diagnosis and treatment participating in the DiTECT-HAT-WP2 study in DR Congo.** SSS: serological screening sites; CDT centre for diagnosis and treatment. The base map layer (left map) was obtained from the Database of Global Administrative Area GADM (https://geodata.ucdavis.edu/gadm/gadm4.1/shp/gadm41_COD_shp.zip) under the license https://gadm.org/license.html. The health zones layer (right map) from Humanitarian Data Exchange (https://data.humdata.org/dataset/zones-de-sante-rdc) under the license https://data.humdata.org/faqs/licenses. The figure was created using http://qgis.org/en/site/.

posts or centers of Dingi Dingi, Kikimi, Kimbuala, Liboke, Mama Marie, Menkao, Mokali, Ngamanzo and Sophora, and the hospital of Mbankana. The SSS in Kwilu were health posts or centers of Bagata 1, Ebay, Kama, Kisakinda, Lumbu, Masamuna, Mpene I, Musaba, Mushie Pentane, Saint Joseph and Saint Paul II Etna. In Kinshasa, the Maluku and Roi Baudoin hospitals acted as CDT, in Kwilu the health reference centres and hospitals of Bandundu, Bangumi, Bagata, Nkara, Masamuna and Mokala.

## Study protocol

The study protocol and tests carried out were the same as in Côte d'Ivoire and Guinea [12,13]. The study documents and all standard operating procedures of tests mentioned below are deposited in https://doi.org/10.23708/LTTOWL. Inclusion criteria were presence in a HAT endemic area; and presenting with clinical suspicion for HAT (recurrent fever not responding to anti-malarial medication; persistent headache; enlarged cervical lymph nodes; important weight loss; weakness; severe itching; amenorrhea, abortion, or sterility; coma; psychiatric problems; sleep perturbation; motor disorders; or

speech disorders). Exclusion criteria were having been treated previously for HAT; absence of written informed consent (and assent for minors); or being less than 4 years old.

Finger prick blood from participants was tested with the HAT Sero *K*-SeT (Coris Bioconcept, Belgium), rHAT Sero-Strip (Coris Bioconcept, Belgium) and SD Bioline HAT (Abbott, South Korea) RDTs. HAT RDT negative participants were considered HAT free, while participants positive in at least one RDT were considered serological suspects and referred for parasitological examination. If lymphadenopathy was present, lymph was collected and microscopically examined. If no lymphadenopathy was present or lymph was parasite negative, 4 ml of heparinised blood was examined using the mini anion exchange centrifugation technique on buffy coat (mAECT-BC) [17]. The cerebrospinal fluid (CSF) of parasito-logically confirmed HAT patients and of RDT seropositives with strong clinical suspicion was examined for cytorachia and for presence of trypanosomes [18]. Treatment of HAT was carried out according to the protocols in place. Unconfirmed serological suspects were re-examined by the CDT or by the national sleeping sickness program (PNLTHA).

For HAT RDT positives, dried blood spots (DBS) were prepared for further remote analysis in the reference laboratory [12,13]. Drops of 30 µl of heparinised blood were spotted on Whatman grade 4 filter paper. In parallel, 180 µls of heparin-ised blood were lysed for 5 minutes with 20 µls of 5% SDS (Sigma Aldrich), and 2x40 µls of lysed blood were deposited on Whatman grade 1001 filter paper.

## Reference laboratory tests

The DBS were sent to the WHO collaborating centre for reference and training on diagnosis of human African trypano-somiasis at the Institut National de Recherche Biomédicale (INRB, Kinshasa, DRC) for further serological and molecular testing, as previously described [12,13]. The DBS on Whatman grade 4 paper were analysed for *T.b. gambiense* specific antibodies by trypanolysis and indirect ELISA/*T.b. gambiense* and for *Trypanozoon* DNA by m18S qPCR and TgsGp-qPCR. The lysed blood on Whatman grade 1001 was tested with Loopamp *Trypanosoma brucei* Detection Kit (LAMP, Eiken Chemical, Tokyo, Japan). Standard Operation Procedures are available at https://doi.org/10.23708/LTTOWL.

## Data analysis

Results were entered in a digital case report form (CRF) [19] and exported into a Microsoft Excel sheet. Participants who were RDT positive, but did not undergo parasitological examination, or had missing or inconsistent basic data (e.g., non-eligibility; miscoding; incomplete RDT results; or missing parasitological confirmation despite RDT positivity) were excluded from the analysis. Descriptive statistics were carried out. Only participants with microscopically demonstrated trypanosomes were considered HAT positive. Participants that were RDT negative, or in whom no trypanosomes could be detected after microscopic examination(s), were considered HAT negative. The participants HAT status was used as gold standard. Associations of clinical symptoms and signs with the HAT status were assessed by regression analysis (Statistical Package: SPSS) in order to identify, among the 13 clinical presentations used for inclusion in the study, which ones are most suited to better target RDT testing for HAT diagnosis in passive screening. Evaluation of the diagnostic performance was also based on the participants HAT status (Fig 2). The diagnostic performance of the clinical presentation (only those that were retained in mixed logistic regression), of the three RDTs and of the four laboratory tests was determined: The sensitivity (number of test positives in the HAT test population, over the total HAT test population), specificity (number of test negatives in the non-HAT test population, over the total non-HAT test population), positive predictive value (PPV; number of test positives in the HAT test population, over the total number of test positives) negative predictive value (NPV; number of test negatives in the non-HAT test population over the total number of test negatives) and diagnostic accuracy (number of test positives in the HAT test population plus number of test negatives in the non-HAT test population over the total number tested) for diagnosis of HAT were calculated [20], with 95% Wilson-Brown confidence intervals (Graphpad Prism 10.4.1). Test agreement was assessed by calculating kappa [21].

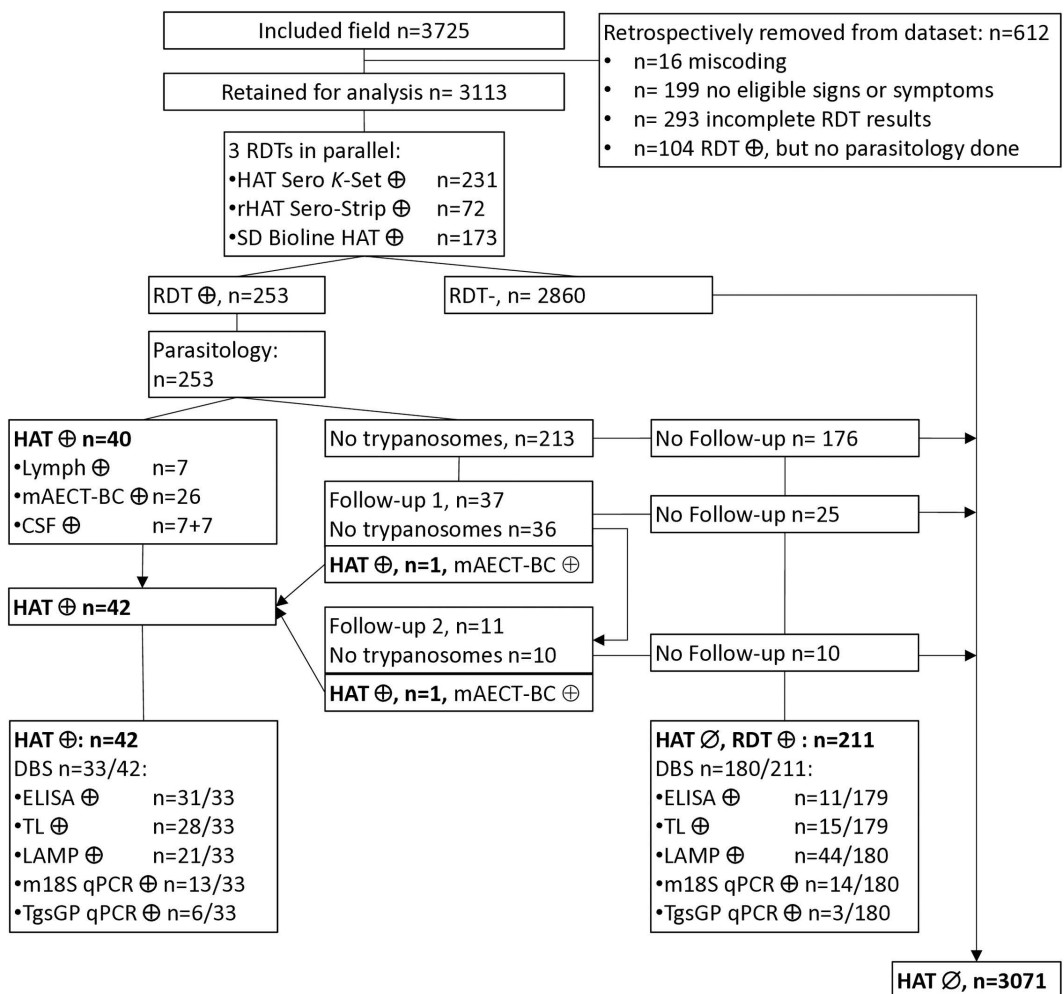

**Fig 2. Flow chart of DiTECT-HAT-WP2 study participants.** Overall results of the rapid diagnostic tests and the reference laboratory tests are indicated and classification as HAT or non-HAT given. RDT: rapid diagnostic test; HAT: human African trypanosomiasis; mAECT-BC: mini anion exchange centrifugation technique on buffy coat; CSF: cerebrospinal fluid; DBS: dried blood spot; ELISA: indirect ELISA/*T.b. gambiense*; TL: trypanolysis; LAMP: Loopamp *Trypanosoma brucei* Detection Kit.

## Results

### Descriptive analysis

Although 3725 clinical suspects were initially included, 612 (16.4%) needed to be retrospectively excluded (Fig 2) from the analysis due to: miscoding (n = 16); non-eligibility (n = 199 did not have signs or symptoms allowing their inclusion); incomplete RDT results (n = 293); or missing parasitological confirmation despite positivity in ≥ 1 RDT (n = 104). Study results from 3113 participants were therefore analysed (Fig 2). In the province of Kinshasa 618/3113 (19.9%) participants were recruited, in the province of Kwilu 2495/3113 (80.1%). In total 1478/3113 (47.5%) participants were included in an SSS, and 1635/3113 (52.5%) in a CDT. Most participants were female 1693/3113 (54.4%), the median age was 29 years (interquartile range: 19–43). The most frequent clinical presentations were headache (61.8%) and recurrent fever (61.2%), followed by weight loss (25.9%), and weakness (25.5%) (Table 1).

**Table 1. Clinical symptoms and signs in study participants and in confirmed HAT patients, and association to HAT positivity. Univariate analysis was performed first to assess the association of clinical presentation to HAT positivity, followed by multivariate analysis. OR: odds ratio; CI: confidence interval.**

| Variable | All participants N = 3113 Frequency (number) | HAT N = 42 Frequency (number) | Univariate analysis P value | OR [95% CI] | Mixed logistic regression P value | OR [95% CI] |
|---|---|---|---|---|---|---|
| Sex (Male) | 45.6% (1420) | 66.7% (28) | 0.008 | 2.4 [1.3-4.6] | 0.01 | 2.6 [1.2-5.2] |
| Sleep disruption | 10.9% (338) | 61.9% (26) | <0.001 | 14.4 [7.6-27.1] | <0.001 | 10.6 [5.4-21.2] |
| Enlarged lymph node | 7.4% (229) | 40.5% (17) | <0.001 | 9.2 [4.9-17.2] | <0.001 | 7.0 [3.3-14.9] |
| Psychiatric problems | 6.0% (187) | 21.4% (9) | <0.001 | 4.4 [2.1-9.4] | 0.001 | 4.5 [1.8-11.0] |
| Motor disorders | 8.7% (271) | 21.4% (9) | 0.005 | 2.9 [1.4-6.2] | 0.015 | 3.0 [1.2-7.4] |
| Weakness | 25.5% (795) | 47.6% (20) | 0.001 | 2.7 [1.5- 5.0] | 0.06 | 2.0 [1.0-4.2] |
| Weight loss | 25.9% (806) | 47.6% (20) | 0.002 | 2.6 [1.4- 4.9] | 0.3 | 1.5 [0.7-3.1] |
| Itching | 12.6% (392) | 26.2% (11) | 0.01 | 2.5 [1.2-5.0] | 0.1 | 1.9 [0.8-4.3] |
| Recurrent fever | 61.2% (1905) | 78.6% (33) | 0.02 | 2.3 [1.1-4.9] | 0.004 | 3.3 [1.5-7.3] |
| Headache | 61.8% (1924) | 69.0% (29) | 0.3 | 1.4 [0.7-2.7] | | |
| Speech disorders | 2.6% (81) | 2.4% (1) | 0.9 | 0.9 [0.1-6.7] | | |
| Coma | 0.6% (18) | 0.0% (0) | 1.0 | 0 | | |
| Amenorrhea* | 5.5% (65/1186) | 0.0% (0) | 1.0 | 0 | | |
| Convulsions | 4.6% (142) | 0.0% (0) | 1.0 | 0 | | |

* only females from 14-49 years old, numbers are shown between brackets

Overall, 253/3113 study participants were positive in at least one HAT RDT (8.1%; 95% CI: 7.2- 9.1%), including 231/3113 in HAT Sero *K*-SeT (Fig 2). Among those 253 RDT positives, in total 42 individuals were confirmed as HAT patient by detection of trypanosomes, giving a HAT prevalence of 1.35% (42/3113). Parasitological confirmation was immediate -upon the first parasitological examination- for 40 HAT patients, of which 7 in lymph (not tested anymore in mAECT-BC) and 26 in mAECT-BC. Examination of the CSF was parasite positive in 14 HAT patients (out of 20 with CSF examined for trypanosome presence), of which 7 had not been previously confirmed by lymph or blood examination. One HAT patient was confirmed after a second parasitological examination at the first follow up visit 7 months post-inclusion (mAECT-BC positive), and another one after a third parasitological examination at the 2nd follow-up 17 months after inclusion (mAECT-BC positive). The median CSF white blood cell count of the HAT patients was 244/µl (n = 22/42, range: 3–2471/µl), among the 22 HAT patients who underwent staging, there was only one stage 1 patient. 2860 RDT negatives, as well as 213 RDT positives who had one (n = 176), two (n = 25) or 3 (n = 10) parasite negative examinations (mainly mAECT-BC), were considered HAT negative (Fig 2). The HAT prevalence observed in passive screening in the DiTECT-HAT WP2 study participants was 1.35% (42/3313; 95% CI 1.00-1.82).

## Clinical presentation: association with HAT and diagnostic performance

Table 1 shows the frequency of the clinical signs and symptoms in confirmed HAT cases. After logistic regression, sleep disruption, enlarged lymph nodes, psychiatric problems, recurrent fever not responding to anti-malarial medication and motor disorders were associated with HAT. Sleep disruption and enlarged lymph nodes had the highest odds ratios, of 10.6 and 7.0. Man were more likely to suffer from sleeping sickness than woman.

The presence of one of the 5 clinical signs retained by logistic regression (sleep disruption, enlarged lymph nodes, psychiatric problems, motor disorders or fever in parallel) had 95.2% sensitivity (40/42; 95% CI 84.2-99.2%), but only 22.8% specificity (700/3071; 95% CI 21.3-24.3%) for HAT, mainly due to the high frequency of recurrent fever. When disregarding fever, presence of sleep disruption, enlarged lymph nodes, psychiatric problems or motor disorders had 83.3% sensitivity

(35/42; 95% CI 69.4-91.7%) and 72.3% specificity (2221/3071; 95% CI 70.7-73.9%) for HAT. As HAT patients with motor disorders always had one of the other 3 signs, the presence of enlarged lymph nodes, sleep disruption or psychiatric problems had a sensitivity of 83.3% (35/42; 95% CI 69.4-91.7%) for HAT, whereas specificity increased to 79.1% (2428/3071; 95% CI 77.6-80.5%). Positive and negative predictive values for HAT (PPV and NPV) of presence of one of these 3 signs were respectively 5.2% (35/678; 95% CI 3.7-7.1%) and 99.7% (2428/2435; 95% CI 99.4-99.9%). Presence of enlarged lymph nodes or sleep disruption had 78.6% sensitivity (33/42; 95% CI 64.1-88.3%), 83.8% specificity (2573/3071; 95% CI 82.4-85.1%), 6.2% PPV (33/531; 95% CI 4.4-8.6%) and 99.7% NPV (2573/2582; 95% CI 99.3-99.8%) for HAT.

## Performance of diagnostic tests

Details of the diagnostic performances of the RDTs applied in passive screening are shown in Table 2. HAT Sero *K*-SeT had 100% diagnostic sensitivity, but presented the lowest specificity (93.9%) of the 3 RDTs that were applied. Its PPV and NPV were respectively 18.2 and 100%, its diagnostic accuracy was 93.9% (2924/3113; 95% CI 93.0-94.7%).

Among the 253 RDT positives, DBS were available for 213 individuals, including 33 HAT patients and 180 HAT negatives. Results of the remote reference laboratory tests carried out on DBS are shown in Table 3 (immunological tests missing for 1). Sensitivities of the ELISA/*T.b. gambiense* and trypanolysis were respectively 93.9 and 84.9%, specificities 93.9 and 91.6%. Parallel combination of these immunological tests gave a similar sensitivity of 93.3% (31/33; 95% CI 80.4-98.9%), and slightly decreased specificity to 90.5% (162/179; 95% CI 85.3-94.0%), with PPV and NPV of respectively 64.6% (31/48; 95% CI 50.4-76.6%) and 98.8% (162/164; 95% CI 95.7-99.8%). Test agreement between ELISA/*T.b. gambiense* and trypanolysis was almost perfect with a kappa of 0.84 (95% CI 0.75-0.93).

The diagnostic sensitivities of the molecular tests were 63.6% for LAMP, 39.4% for m18S qPCR and 18.2% for TgsGp qPCR (Table 3). Specificities were respectively 75.6, 92.2 and 98.3%. Parallel combination of the 3 molecular tests gave a sensitivity of 75.8% (25/33; 95% CI 59.0-87.2%) and specificity of 71.1% (128/180; 95% CI 64.1-77.2) with PPV and NPV of respectively 32.5% (25/77; 95% CI 23.1-43.5) and 94.1% (128/136; 95% CI 88.8-97.0). Test agreement between LAMP and m18S qPCR, which are both *Trypanozoon* specific was fair (kappa of 0.21; 95% CI 0.07-0.34), between m18S qPCR

**Table 2. Diagnostic performance of 3 rapid diagnostic tests for passive screening of human African trypanosomiasis in DR Congo. Sensitivity, specificity, positive predictive value (PPV) and negative predictive value (NPV) were calculated with 95% confidence intervals (95% CI).**

|  | % Sensitivity (n/N; 95% CI) | % Specificity (n/N; 95% CI | % PPV (n/N; 95% CI | % NPV (n/N; 95% CI |
|---|---|---|---|---|
| HAT Sero *K*-SeT | 100 (42/42; 91.6-100) | 93.9 (2882/3071; 92.9-94.6) | 18.2 (42/231; 13.7-23.7) | 100 (2882/2882; 99.9-100) |
| rHAT Sero-Strip | 61.9 (26/42; 0.47-0.75) | 98.5 (3025/3071; 98.0-98.9) | 36.1 (26/72; 26.0-47.7) | 99.5 (3025/3041; 99.2-99.7) |
| SD Bioline HAT | 90.5 (38/42;77.9-96.2) | 95.6 (2936/3071; 94.8-96.3) | 22.0 (38/173;16.4-28.7) | 99.9 (2936/2940; 99.7-100) |

**Table 3. Diagnostic performance of remote reference laboratory tests on dried blood spots for human African trypanosomiasis in DR Congo. Sensitivity, specificity, positive predictive value (PPV) and negative predictive value (NPV) were calculated with 95% confidence intervals (95% CI).**

|  | % Sensitivity (n/N; 95% CI) | % Specificity (n/N; 95% CI) | % PPV (n/N; 95% CI) | % NPV (n/N; 95% CI) |
|---|---|---|---|---|
| Immunological tests |  |  |  |  |
| ELISA/*T.b. gambiense* | 93.9 (31/33; 80.4-98.9) | 93.9 (168/179; 89.3-96.5) | 73.8 (31/42; 58.9-84.7) | 98.8 (168/170; 95.8-99.8) |
| Trypanolysis | 84.9 (28/33; 69.1-93.4) | 91.6 (164/179; 86.6-94.9) | 65.1 (28/43; 50.2-77.6) | 97.0 (164/169; 93.3-98.7) |
| Molecular tests |  |  |  |  |
| LAMP | 63.6 (21/33; 46.6-77.8) | 75.6 (136/180; 68.8-81.3) | 32.3 (21/65; 22.2-44.4) | 91.9 (136/148; 86.4-95.3) |
| m18S qPCR | 39.4 (13/33; 24.7-56.3) | 92.2 (166/180; 87.4-95.3) | 48.2 (13/27; 30.7-66.0) | 89.3 (166/186; 84.0-92.9) |
| TgsGp qPCR | 18.2 (6/33; 8.6-34.4) | 98.3 (177/180; 95.2-99.6) | 66.7 (6/9;35.4-87.9) | 86.8 (177/204; 81.4-90.7) |

and TgsGp qPCR was moderate (kappa of 0.60; 95% CI 0.44-0.75), and between LAMP and TgsGp qPCR was slight (kappa of 0.04; 95% CI 0.05-0.12).

All DBS from HAT patients were positive in at least 1 reference laboratory test (100% sensitivity; 33/33; 95% CI 89.6-100.0%), and 69.7% in at least one molecular and one immunological test (23/33; 95% CI 52.7-82.6). Among the non parasitologically confirmed RDT seropositives, 63/179 DBS were positive in at least 1 reference laboratory test (64.8% specificity; 95% CI 57.6-71.4), and 5/179 in at least one molecular and one immunological test (97.2% specificity; 93.6-98.8%). One person, not considered as a HAT patient, was positive for all 5 reference laboratory tests. The PPV and NPV of serial combination of at least one molecular and one immunological test were respectively 82.1% (23/28; 95% CI 64.4-92.1%) and 94.6% (174/184; 95% CI 90.3-97.0%). Among the 5 parasitologically negative RDT positives that were both positive in a molecular and a serological test, unfortunately none underwent an additional parasitological examination.

## Discussion

The present study was undertaken to assess the diagnostic performance of the clinical presentation, of 3 HAT RDTs and of reference immunological and molecular laboratory tests in a context of passive screening. The study took place in health zones in Kwilu and Kinshasa province in DR Congo, which are among the ones with the highest HAT prevalence in Africa.

Among the clinical presentations selected from literature [22,23] and serving as inclusion criteria for the study, sleep disruption, enlarged lymph nodes, psychiatric problems, recurrent fever and motor disorders were significantly associated to HAT. Presence of either swollen lymph nodes, sleep disruption and psychiatric problems had odds higher than 4 and presence of at least one of these signs had about 80% sensitivity and specificity, and respectively 5.2% and 99.7% PPV and NPV. Swollen lymph nodes and sleep disorders are characteristic for "sleeping sickness", and well known in HAT endemic areas [23,24]. Sleep disorders and change in behaviour had, in another diagnostic study in the same region, already been observed as the most frequent clinical signs occurring in HAT patients [14].

The sensitivity of the three HAT RDTs in DR Congo was similar to their sensitivities observed in the DiTECT-HAT-WP2 study in Guinea [12]. In particular for HAT Sero *K*-SeT, 100% or nearly 100% sensitivity was previously observed in the Kwilu province [14,15]. The RDT specificities observed in the present study appeared to be slightly lower than in Côte d'Ivoire and Guinea [12,13]. For HAT Sero *K*-SeT, the 93.9% specificity observed presently was also slightly lower than HAT Sero *K*-SeT specificities observed in previous studies in Kwilu of 98.6% in active screening [15] and 97.0% in passive screening [14], when HAT prevalence was still higher. On the other hand, HAT Sero *K*-SeT specificity was slightly higher than the 86.7% and 91% specificities with this test observed in more recently active screening studies in respectively West-Africa [10] and DR Congo (NCT05637632, Tablado, personal communication). Compared to the target product profile established by the World Health Organization for a HAT screening test to identify individuals to receive widened treatment [5], the diagnostic sensitivity of the HAT Sero *K*-SeT appears ideal, whereas specificity seems to be rather around minimally acceptable or below, depending on the study and screening method.

Due to their high specificity, trypanolysis and indirect ELISA/*T.b. gambiense* on DBS are already quite well established as remote reference laboratory tests for HAT [10,12,13,25–27]. Their sensitivities on DBS are however less documented. With values of respectively 85.3 and 67.6%, sensitivity of trypanolysis and of indirect ELISA/*T.b. gambiense* in DiTECT-HAT-WP2 in Guinea turned out to be lower than expected [12]. With 84.9% sensitivity, the actual study confirms this moderate sensitivity of trypanolysis on DBS in Guinea. With 93.9% sensitivity in the actual study, indirect ELISA/*T.b. gambiense* on DBS in DR Congo seems to perform better than in Guinea, and better than the 82.2% sensitivity estimated earlier in DR Congo [25]. Diagnostic performances of LAMP, m18S qPCR and TgsGp qPCR on DBS observed in DiTECT-HAT-WP2 DR Congo were in line with observations in Guinea [12]. These molecular tests had overall low diagnostic sensitivity while specificities were over 90%, except for LAMP in DR Congo. Despite the low reference laboratory test sensitivities, all DBS of HAT patients were at least positive in one of the 5 tests (100% sensitivity), and serial combination

of positivity in at least one molecular and one immunological test had 97.2% specificity. The former criterion can therefore be applied when maximum sensitivity is required, the latter when maximum specificity is required. As parasitology was used as a gold standard and sensitivity of parasitological tests is less than 100%, we cannot entirely exclude that the 5 RDT positive individuals which could not be confirmed parasitologically but were positive in both an immunological and a molecular reference test, were real HAT patients, in particular the one individual that was positive in all 5 reference laboratory tests carried out.

This study in DR Congo had some limitations. Some of these are inherent to the DiTECT-HAT-WP2 study design and have been discussed elsewhere [12,13]: non-inclusion of individuals without the selected clinical signs and symptoms, imperfect gold standard, inclusion of almost exclusively 2nd stage HAT patients in passive screening, some subjectivity in assessing clinical symptoms and signs and only fragmentary follow-up for parasite negative RDT positives, even for those with positive reference laboratory tests. Due to the lack of parasitological examination of all RDT negatives, and the fact that the evaluated RDTs largely rely on the same VSG antigens (complemented with ISG65 for rHAT Sero-Strip), we cannot exclude a risk of overestimation of sensitivity. Carrying out parasitology on all RDT negative participants would have been unfeasible in this field based prospective study, due to the additional work load, but also because of the lack of confirmation capacity in the SSS which would have obliged all participants to go to the CDT, or to limit the study to CDTs only. In the present study, in particular the retrospective exclusion of 16.4% of the study participants is an important weakness, and shows the challenges of conducting a multi-centre diagnostic trial with most health staff having little or no clinical trial experience, in hard to reach remote settings where HAT typically occurs. Of the excluded individuals, about half (47.9%, 293/612) had a missing RDT result, which was a consequence of the difficult logistics that had to be dealt with. Also, 104/357 (29.1%) RDT positives missed results of parasitological examination. This proportion was much higher than in Guinea in the same study, probably also due to the larger distances and limited transport options in rural DR Congo. This observation underlines the shortcomings of multi-step HAT diagnosis with lack of confirmation capacity in SSS. A number of potential HAT patients might not have reached parasitological examination in the CDT or might have gone to a CDT not included in this study. We suspect that the majority of these were probably false RDT positives or stage 1 patients with relatively non-severe symptomatology, as it is assumed that for individuals with neurological stage HAT symptoms, the probability for referral is higher as they feel sick. The loss of HAT patients along the diagnostic pathway during active screening was previously estimated to be almost 50% of the patients, who would not receive treatment as a result [28]. These numbers also demonstrate the importance and potential impact of a future "screen and treat" strategy for elimination of HAT [6]. Finally, only 1 out of the 3 RDTs evaluated in the DiTECT-HAT-WP2 study is still commercialised. The Abbott Bioline HAT 2.0, which is now commercialised and of which specificity, but not sensitivity, has recently been tested prospectively [10], was unavailable.

Despite its weaknesses, the DiTECT-HAT-WP2 study in DR Congo also has its strengths. Specificity data are based on a large number of study participants. Also, due to the HAT prevalence in the study zone, which was higher than in most other endemic countries, the sensitivity of the HAT RDTs and of the reference laboratory tests used to remotely confirm HAT, could be documented further.

From a clinical point of view, the study results have several implications. First, patients presenting with swollen lymph nodes, sleep disruption or psychiatric problems presenting at health facilities in HAT endemic zones in DR Congo, should undergo an RDT to screen for HAT. The study allowed to confirm the excellent sensitivity of the HAT Sero *K*-SeT RDT, under field conditions, but highlights again the lack of specificity of the actual RDTs. The loss of RDT positive individuals, between serological screening and parasitological confirmation, underlines the potential of a "screen and treat" strategy with a safe and easy to use drug for elimination of HAT [6]. On the other hand, the suboptimal specificity of the HAT Sero *K*-SeT RDT might in a "screen and treat" strategy, result in overtreatment. Furthermore, the results confirm sub-optimal sensitivity of both trypanolysis and to a lesser extent, indirect ELISA/*T.b. gambiense*, carried out on dried blood spots, underlining the risk of missing true HAT positives when applying only one of these tests, or applying them in serial. Our

results also underline the need to improve molecular tests for remote testing. New storage systems with DNA/RNA stabilisation buffer and new molecular tests, including a *Trypanozoon* S$^2$ RT-qPCR allow RNA and DNA conservation and detection, and are being evaluated further [6,10,29]. Despite the imperfections of the individual reference laboratory tests, parallel or serial combination of immunological and molecular test positivity, can be used to select either for maximum sensitivity and identify individuals at risk for HAT who merit to be followed-up further, or for maximum specificity, for epidemiological purposes. This principle is already applied as a remote test algorithm in the STROGHAT screen and treat study (NCT06356974) [6]. However, overall, better diagnostics, compliant to the respective target product profiles established by WHO [5,30], remain required to support elimination of HAT.

## Supporting information

**S1 File. Completed STARD checklist for abstracts.**
(DOCX)

**S2 File. Completed STARD checklist.**
(DOCX)

## Acknowledgments

We would like to thank Albertine Mukolo and Jean Roger Kalo Lilo for their assistance with training. All staff of the SSS and CDT participating in the study are acknowledged for their work, as well as the HAT clinical suspects that agreed to participate. We thank Massimo Paone from FAO for creating the map.

## Author contributions

**Conceptualization:** Crispin Lumbala, Philippe Büscher, Dieudonné Mumba Ngoyi, Veerle Lejon.

**Data curation:** Jacquies Makabuza.

**Formal analysis:** Jacquies Makabuza, Veerle Lejon.

**Funding acquisition:** Crispin Lumbala, Philippe Büscher, Dieudonné Mumba Ngoyi, Veerle Lejon.

**Investigation:** Jacquies Makabuza, Ipos Ngay Lukusa, Crispin Lumbala, Erick Mwamba Miaka, Pathou Nganzobo, Alain Fukinsia, Jean Kwete, Nicolas Bebronne.

**Methodology:** Jacquies Makabuza, Ipos Ngay Lukusa, Jean Kwete, Nicolas Bebronne.

**Project administration:** Crispin Lumbala, Erick Mwamba Miaka, Philippe Büscher, Dieudonné Mumba Ngoyi, Veerle Lejon.

**Supervision:** Jacquies Makabuza, Crispin Lumbala, Erick Mwamba Miaka, Pathou Nganzobo, Alain Fukinsia, Philippe Büscher, Dieudonné Mumba Ngoyi, Veerle Lejon.

**Validation:** Jacquies Makabuza, Philippe Büscher.

**Writing – original draft:** Jacquies Makabuza, Veerle Lejon.

**Writing – review & editing:** Ipos Ngay Lukusa, Crispin Lumbala, Erick Mwamba Miaka, Pathou Nganzobo, Alain Fukinsia, Jean Kwete, Nicolas Bebronne, Philippe Büscher, Dieudonné Mumba Ngoyi.

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
