## [Decision Letter · Decision Letter 0]

3 Jun 2025

PNTD-D-25-00537Passive surveillance of human African trypanosomiasis in the Democratic Republic of the Congo: clinical presentation and prospective evaluation of rapid diagnostic and reference laboratory test accuracy. PLOS Neglected Tropical Diseases Dear Dr. Lejon, Thank you for submitting your manuscript to PLOS Neglected Tropical Diseases. After careful consideration, we feel that it has merit but does not fully meet PLOS Neglected Tropical Diseases's publication criteria as it currently stands. Therefore, we invite you to submit a revised version of the manuscript that addresses the points raised during the review process. Please submit your revised manuscript within 30 days Aug 02 2025 11:59PM. If you will need more time than this to complete your revisions, please reply to this message or contact the journal office at plosntds@plos.org. Please include the following items when submitting your revised manuscript: * A rebuttal letter that responds to each point raised by the editor and reviewer(s). You should upload this letter as a separate file labeled 'Response to Reviewers'. This file does not need to include responses to any formatting updates and technical items listed in the 'Journal Requirements' section below. * A marked-up copy of your manuscript that highlights changes made to the original version. You should upload this as a separate file labeled 'Revised Manuscript with Track Changes'. * An unmarked version of your revised paper without tracked changes. You should upload this as a separate file labeled 'Manuscript'. If you would like to make changes to your financial disclosure, competing interests statement, or data availability statement, please make these updates within the submission form at the time of resubmission. Guidelines for resubmitting your figure files are available below the reviewer comments at the end of this letter.

We look forward to receiving your revised manuscript.

Kind regards,

Epco Hasker

Academic EditorPLOS Neglected Tropical Diseases

Guilherme Werneck

Section Editor

Shaden Kamhawi

co-Editor-in-Chief

Paul Brindley

co-Editor-in-Chief

**Journal Requirements:**

1) Please upload figure 1 as a separate Figure file in .tif or .eps format. For more information about how to convert and format your figure files please see our guidelines: 

2) We have noticed that you have uploaded Supporting Information files, but you have not included a list of legends. Please add a full list of legends for your Supporting Information files after the references list.

3) Some material included in your submission may be copyrighted. According to PLOSu2019s copyright policy, authors who use figures or other material (e.g., graphics, clipart, maps) from another author or copyright holder must demonstrate or obtain permission to publish this material under the Creative Commons Attribution 4.0 International (CC BY 4.0) License used by PLOS journals. Please closely review the details of PLOSu2019s copyright requirements here: PLOS Licenses and Copyright. If you need to request permissions from a copyright holder, you may use PLOS's Copyright Content Permission form.

Potential Copyright Issues:

i) Figure 1. Please (a) provide a direct link to the base layer of the map (i.e., the country or region border shape) and ensure this is also included in the figure legend; and (b) provide a link to the terms of use / license information for the base layer image or shapefile. We cannot publish proprietary or copyrighted maps (e.g. Google Maps, Mapquest) and the terms of use for your map base layer must be compatible with our CC BY 4.0 license.

**Reviewers' comments:**

Reviewer's Responses to Questions

**Key Review Criteria Required for Acceptance?**

**Methods**

-Are the objectives of the study clearly articulated with a clear testable hypothesis stated?

-Is the study design appropriate to address the stated objectives?

-Is the population clearly described and appropriate for the hypothesis being tested?

-Is the sample size sufficient to ensure adequate power to address the hypothesis being tested?

-Were correct statistical analysis used to support conclusions?

-Are there concerns about ethical or regulatory requirements being met?

Reviewer #1: In the context of declining prevalence and the WHO's proposed SPTs, we feel that this study has clear objectives, and by using parasitology as the gold standard for confirming cases of HAT, the RDTs have been correctly evaluated.

The sample size could not be very large given the drop in prevalence, but selection on the basis of clinical suspicion is what is recommended.

To compensate for the small sample size, the multicentricity of the study provided sufficient power.

The statistical analyses used were sound, and ethical requirements were met.

Reviewer #2: The methods section describes accurately how the study was conducted, including information on the study protocol and the methods for data analysis. For the diagnostic assays, except for the RDTs, there is only a mention in the methods without detailed description whether they were produced in house or if they were supplied by a company. A more detailed description is required for the assays listed under “trypanolysis and indirect ELISA/T.b. gambiense and for Trypanozoon DNA by m18S qPCR and TgsGp-qPCR”.

Reviewer #3: Under the data analysis section, the authors show that the specificity and sensitivity was analysed in graphpad, but its not clear at what level this was done. What was the gold standard for this and why and which test kit comparisons were done in comparison to the gold standard. More details need to be provided here. It is also not clear what specificity means in this case as suspected patients with negative parasitological outcome were excluded from the analysis, more information on what formed the specificity analysis needs to be provided.

The authors looked at clinical presentation and RDT testing in line with parasitilogical examination and molecular based testing. However, I do not see any relationship between the factors being studied. Why did the authors look at clinical presentation when they are testing diagnostic tests. My thinking was that they wanted to associate key clinical pictures with the diagnostic test outcomes to see if there is a clinical picture that could predict test outcome, eg do individuals who have a certain symptom/symptoms or a set of them more likely to be positive with a certain test and thus clinician can use this clinical picture to treat without parasitological confirmation. I would have loved to see this type of analysis. line 223 seems to be about this but on deeper reading it is not and instead (how this analysis for this section was done is not indicated in the data analysis section of the methods)

**Results**

-Does the analysis presented match the analysis plan?

-Are the results clearly and completely presented?

-Are the figures (Tables, Images) of sufficient quality for clarity?

Reviewer #1: In reading the results, we note that the analyses made are based on the statements made in the plan; the various tables contribute to the understanding and synthesis of the quantitative results.

Data exclusions from the analysis have been justified.

Reviewer #2: The description of the results is clear and easy to follow. I just recommend the authors carefully check how the data is presented in the tables. For example, I could not figure out why in Table 1 there is a line named “Sex male” but no line for female.

Reviewer #3: • I would have loved to see results on how the clinical picture relates to the diagnostic kit performance.

• The tables need to be improved for clarity in terms of the design and content. The column for the tests needs to show this and headings should be bolded for clarity

• Line 246. Am concerned about the 18.2% of the PPV of the HAT Sero K-SeT. If this is true then the authors need to check the analysis, otherwise it appears not to add up.

• Line 246, talks about diagnostic accuracy, I am not sure how this was calculated and needs to be indicated in the data analysis section of the paper.

**Conclusions**

-Are the conclusions supported by the data presented?

-Are the limitations of analysis clearly described?

-Do the authors discuss how these data can be helpful to advance our understanding of the topic under study?

-Is public health relevance addressed?

Reviewer #1: Yes, the conclusions are in line with the data presented, while noting the weaknesses due to the sample size, and the sensitivity and specificity of the various tests used.

The authors have shown that these RDTs, although having limitations, can be used especially in the test-and-treat strategy with non-toxic drugs and may contribute to the elimination of HAT, and they have recommended continuing studies with these RDTs in combination to increase sensitivity and to have a large sample size for subsequent evaluations.

Reviewer #2: The findings as very relevant concerning both the epidemiology aspects and also the assessment of the performance of the diagnostic tests. A significant number of individuals was tested providing an indication on the level of HAT prevalence in the regions comprised in the study. In addition, the paper raises questions about the usefulness of the molecular tests for remote testing since their performance was very poor. The authors try to be positive, concluding that by combining different tests more conclusive data can be generated. However, they can emphasize in a stronger way that better tests are urgently needed.

Reviewer #3: The authors have tried to discuss the limitation of their study and this seems to make the discussion strong. The conclusions on molecular tests need to be improved and evaluated appears far fetched as the study is looking at clinical presentations and rapid diagnostic tests, for which no clear conclusion has been provided to this effect, see line 46

**Editorial and Data Presentation Modifications?**

Reviewer #1: No suggestions

Reviewer #2: A request for additional information required in the methods section and a review of the tables in the results scection are mentioned above.

Reviewer #3: The authors need to improve the data analysis section of the methods to indicate the different tests for which results are being reported in the result section. the authors report in the results a relationship between the clinical picture and diagnostic outcome but this is not indicated in the analysis plan. i would also have like to see a section relating clinical picture of a set of clinical symptoms to diagnostic kit performance.

**Summary and General Comments**

Reviewer #1: This publication shows the efforts that partners must make to respond to the WHO's TPP in order to find tests that are easy to use in the context of falling prevalence, and above all what must be used to prove the cessation of transmission. There is still a great deal to be done, and researchers are encouraged to adapt even the methodologies to a particular context.

Reviewer #2: The paper describes the efforts to monitor human African trypanosomiasis (HAT) in several locations in the Democratic Republic of the Congo in the context of the WHO-driven effort to eliminate gambiense HAT. Over three thousand people have been tested over a three-year period between October 2017 and December 2020. The paper deals also with the limitations of rapid diagnostic test for monitoring HAT at screening centers and with the low efficiency of molecular tests for remote testing. Although rapid diagnostic tests also need improvement, the extremely low performance of the molecular tests is very concerning. All this information is relevant to be published.

Reviewer #3: The authors describe the clinical presentation and performance of selected RDTs and reference tests on gHAT diagnosis. The study offers promise on the diagnosis of HAT, although some of the kits might not be readily available. Although the study has limited novelty, the power of the study makes it interesting and provides insights how diagnostic kit performance. the weakness relates to the data analysis as described above and this could be improved by the research team.

PLOS authors have the option to publish the peer review history of their article (what does this mean?). If published, this will include your full peer review and any attached files.

Reviewer #1: **Yes: **Dr Augustin Kadima Ebeja

Reviewer #2: No

Reviewer #3: **Yes: **charles Drago kato

**Figure resubmission:** While revising your submission, please upload your figure files to the Preflight Analysis and Conversion Engine (PACE) digital diagnostic tool, https://pacev2.apexcovantage.com/. PACE helps ensure that figures meet PLOS requirements. To use PACE, you must first register as a user. Registration is free. Then, login and navigate to the UPLOAD tab, where you will find detailed instructions on how to use the tool. If you encounter any issues or have any questions when using PACE, please email PLOS at figures@plos.org. Please note that Supporting Information files do not need this step. If there are other versions of figure files still present in your submission file inventory at resubmission, please replace them with the PACE-processed versions.
---

## [Editor Report · Decision Letter 1]

1 Sep 2025

PNTD-D-25-00537R1

Passive surveillance of human African trypanosomiasis in the Democratic Republic of the Congo: clinical presentation and prospective evaluation of rapid diagnostic and reference laboratory test accuracy.

Dear Dr. Lejon,

Thank you for submitting your manuscript to PLOS Neglected Tropical Diseases. After careful consideration, we feel that it has merit but does not fully meet PLOS Neglected Tropical Diseases's publication criteria as it currently stands. In particular there are concerns about the sensitivity estimates of the rapid tests which could be artificially high since all samples included were preselected on these three tests, which also rely on the same antigens. Therefore, we invite you to explicitly address this issue in the limitations section and then submit a revised version of the manuscript that addresses this point.

Please submit your revised manuscript within 60 days Oct 01 2025 11:59PM. If you will need more time than this to complete your revisions, please reply to this message or contact the journal office at plosntds@plos.org. Please include the following items when submitting your revised manuscript:

We look forward to receiving your revised manuscript.

Kind regards,

Epco Hasker

Academic Editor

Guilherme Werneck

Section Editor

Shaden Kamhawi

co-Editor-in-Chief

Paul Brindley

co-Editor-in-Chief

**Additional Editor Comments:**

We would like to accept your manuscript for publication in PLoS NTDs but one issue came up that still needs to be resolved. There are concerns about the methodology, the way the samples have been selected. Ideally all samples should have been tested with a gold standard test, which in this case would be parasitology. But we understand that HAT is a rare disease, that parasitological confirmation tests are cumbersome and time consuming, and that for those reasons it is not feasible to do such tests on all samples in a field based prospective study. However, when samples included are selected based on three screening tests that also (partially) rely on the same antigens, there is a serious risk for bias, in particular for overestimating sensitivity. We kindly request you to explicitly address this in the limitations.

**Journal Requirements:**

**Reviewers' Comments:**

**Figure resubmission:**
---

## [Editor Report · Decision Letter 2]

17 Sep 2025

Dear Dr. Lejon,

We are pleased to inform you that your manuscript 'Passive surveillance of human African trypanosomiasis in the Democratic Republic of the Congo: clinical presentation and prospective evaluation of rapid diagnostic and reference laboratory test accuracy.' has been provisionally accepted for publication in PLOS Neglected Tropical Diseases.

Best regards,

Epco Hasker

Academic Editor

Guilherme Werneck

Section Editor

Shaden Kamhawi

co-Editor-in-Chief

Paul Brindley

co-Editor-in-Chief

---

## [Editor Report · Acceptance letter]

Dear Dr. Lejon,

We are delighted to inform you that your manuscript, "Passive surveillance of human African trypanosomiasis in the Democratic Republic of the Congo: clinical presentation and prospective evaluation of rapid diagnostic and reference laboratory test accuracy.," has been formally accepted for publication in PLOS Neglected Tropical Diseases.

Best regards,

Shaden Kamhawi

co-Editor-in-Chief

Paul Brindley

co-Editor-in-Chief
